# Rethinking 1-bit Optimization Leveraging Pre-trained Large Language Models

## Abstract

1-bit LLM quantization offers significant advantages in reducing storage and computational costs. However, existing methods typically train 1-bit LLMs from scratch, failing to fully leverage pre-trained models. This results in high training costs and notable accuracy degradation. We identify that the large gap between full precision and 1-bit representations makes naive adaptation difficult. In this paper, we introduce a consistent progressive training for both forward and backward, smoothly converting the full-precision weights into the binarized ones. Additionally, we incorporate binary-aware initialization and dual-scaling compensation to reduce the difficulty of progressive training and improve the performance. Experimental results on LLMs of various sizes demonstrate that our method outperforms existing approaches. Our results show that high-performance 1-bit LLMs can be achieved using pre-trained models, eliminating the need for expensive training from scratch.

## 1 Introduction

Large Language Models (LLMs) have shown remarkable performance across a wide range of natural language processing tasks, including machine translation, text generation, sentiment analysis, and more. Their ability to understand and generate human-like text has made them essential tools in various applications, such as virtual assistants, customer service, content creation, and academic research. However, the impressive performance of LLMs comes with substantial computational and storage requirements Zhao et al. (2023), creating challenges for their deployment and scalability, especially in resource-constrained environments.

Quantization has emerged as a key technique to address these challenges by reducing the precision of model parameters, thereby lowering both memory usage and computational overhead during inference Frantar et al. (2022); Lin et al. (2024); Xiao et al. (2023); Shao et al. (2023); Liu et al. (2023). By representing weights with lower bit-widths, quantization enables more efficient storage and faster computations, making it feasible to deploy LLMs on edge devices and in environments where latency and energy efficiency are critical. Despite its effectiveness, traditional quantization methods often find a trade-off between compression rate and model accuracy, leaving room for improvement, especially when targeting extreme compression.

Among quantization techniques, 1-bit quantization stands out as an extreme approach, constraining the weights to binary values, typically {-1, +1}, which results in the highest possible compression rate. Recent advancements in 1-bit LLMs have explored both Post-Training Quantization (PTQ) and Quantization-Aware Training (QAT) methodologies Huang et al. (2024); Li et al. (2024); Chen et al. (2024); Guo et al.; Ma et al. (2024b); Xu et al. (2024); Ma et al. (2024a); Shang et al. (2023). PTQ methods, such as BiLLM and ARB-LLM, quantize pre-trained models with a few calibration data, avoiding the need for extensive retraining. In contrast, QAT methods like BitNet and FBI-LLM train the model from scratch with quantization in mind, often requiring substantial computational resources and large datasets. While these approaches have made significant progress in 1-bit quantization, they typically suffer from notable accuracy degradation compared to full-precision models and may introduce additional inference overhead.

Existing methods relying on training from scratch fail to fully leverage the rich knowledge embedded in pre-trained models Wang et al. (2023); Ma et al. (2024a), leading to higher training costs and less efficient utilization of existing resources. Many of these studies argue that training 1-bit LLMs from scratch leads to more stable results compared to weight inheritance Ma et al. (2024a). However, in

Figure 1: An overview of the BinaryLLM framework. We apply 1-bit quantization to all the linear layers in the transformer blocks, quantizing the weights to $\{-1, +1\}$. The core components of BinaryLLM, shown on the right, include consistent progressive training, dual-scaling compensation and binary-aware initialization. These techniques significantly reduce the training difficulty associated with 1-bit compression of pre-trained LLMs.

many quantization schemes, inheriting full-precision parameters is considered essential Liu et al. (2020), as full-precision models contain more valuable and effective information. Therefore, we argue that existing 1-bit LLMs do not fully capitalize on the knowledge embedded in pre-trained models, resulting in higher training costs and suboptimal accuracy. Given the large parameter sizes and the significant gap between 1-bit and full-precision models in LLMs, we believe that more careful optimizations are necessary when inheriting parameters from pre-trained models in the context of 1-bit quantization.

In this paper, we propose a novel 1-bit LLM quantization paradigm designed to systematically address the limitations of current methods. Considering the large error between full-precision weights and the corresponding 1-bit ones, 1-bit quantization using the sign function leads to severe destruction of the pre-trained weights at the beginning of training. Thus we propose a consistent progressive training method, where the quantization of weights is close to linear function at the beginning of training, and close to the sign function at the end of training. During the training process, the nonlinear state of the progressive quantization function is smoothly transitioned, thus minimizing the quantization error and retain the valuable and effective information of pre-trained models as much as possible. Besides, we further integrates binary-aware initialization and dual-scaling compensation to reduce the training difficulty of progressive training. Specifically, to preserve salient weights, we introduce an end-to-end approach to search for scaling factors that transform pre-trained parameters into a binary-friendly distribution with minimal loss. And we incorporate extra learnable scaling factors to balance quantization error minimization and accuracy compensation. Experimental evaluations on LLMs of various sizes demonstrate that our proposed paradigm outperforms existing 1-bit quantization methods, effectively bridging the gap between quantized and full-precision models.

## 2 RELATED WORK

BiLLM Huang et al. (2024) represents the first post-training quantization approach for 1-bit LLMs. It partitions weights into salient and non-salient categories, applying binary residual approximation and bell-shaped splitting, respectively, to reduce quantization error. Further, ARB-LLM Li et al. (2024) introduces an alternating refined binarization algorithm that progressively updates the binarization parameters. This method further improves the handling of salient weights, leading to notable performance gains. Separately, DB-LLM Chen et al. (2024) proposes to decompose 2-bit quantized weights into two independent binary sets and employs deviation-aware distillation to enhance accuracy. Nevertheless, these methods still suffer from substantial performance degradation compared to full-precision models and show pronounced instability to the latest LLMs Zheng et al. (2025).

**Quantization-aware Training for 1-bit LLMs.** BitNet Wang et al. (2023) proposes quantizing the weights of LLMs into $\{-1, 0, +1\}$ using a train-from-scratch approach, demonstrating the potential of ultra-low-bit quantization. FBI-LLM Ma et al. (2024a) extends this to true 1-bit values $\{-1, +1\}$,

| Models | BitNet Wang et al. (2023) | OneBit Xu et al. (2024) | FBI-LLM Ma et al. (2024a) |
|---|---|---|---|
| #Init | Scratch | Pre-trained | Scratch |
| #Bit | 1.58 | 1 | 1 |
| Train Set | RedPajama | Synthetic data | Amber |
| #Tokens | 100 B | 13.5 B | 1.26 T |
| #Batch | 1 M | 128 | 3.9M |
| #GPU hours | 5.3 k | 1.3 k | 262 k |

Table 1: 1-bit training settings on 7B LLMs.

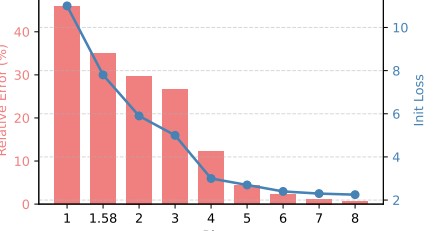

Figure 2: Quantization error and initial loss.

introducing autoregressive distillation to mimic the outputs of full-precision models. However, both methods require vast computational resources and large training datasets to achieve well-performing 1-bit LLMs. In contrast, PB-LLM Shang et al. (2023) and OneBit Xu et al. (2024) construct 1-bit LLMs based on pre-trained parameters. PB-LLM represents some weights with higher-bit precision while binarizing others, incurring excessive inference overhead. And OneBit introduces Sign-Value-Independent Decomposition (SVID) for weight matrices, using approximate 1-bit values and employing quantization-aware knowledge distillation to supervise the training process. But OneBit ignoring the destruction of weights by sign function at the beginning of training.

**Progressive Training for Binary Networks.** Progressive training is an effective method to reduce the quantization error during training. INQ Zhou et al. (2017) proposes to partition the weights into several groups, and gradually converts the full-precision group into low-bit precision group during training. ReActNet Liu et al. (2020) proposed a two-stage training strategy, only quantizing activations in the first stage and then quantizing both weights and activations in the second stage. IR-Net Qin et al. (2020) and RBNN Lin et al. (2020) propose different training-aware approximation function to replace the STE in backward propagation, which gradually approximates the gradient of Sign function as training progresses. However, these methods have only been validated in the quantization of convolutional neural networks (CNNs), which are small and easy to train with limited resources. Besides, These methods use the sign function to quantize the weights in the forward process, which also easily lead to destroying the pre-training parameters of LLMs.

## 3 METHOD

In this section, we firstly demonstrate our motivation of 1-bit training with pre-trained large language models, and then introduce several novel techniques to improve the accuracy of 1-bit LLMs, including progressive training, parameter initialization and dual-scaling compensation.

### 3.1 MOTIVATION

We begin with a brief overview of binary neural networks. Unlike general low-bit quantization methods, which map full-precision values to $b$-bit integers $x_{int} \in [-2^{b-1}, 2^{b-1} - 1]$ using a round-to-nearest function, 1-bit quantization compresses full-precision values by applying a sign function Courbariaux et al. (2016) together with scaling factors Rastegari et al. (2016), as follows.

$$X^b = \mathcal{B}(X) = S_a \times \texttt{Sign}(X), \tag{1}$$

where

$$\texttt{Sign}(X_i) = \begin{cases} +1, & \text{if } X_i \geq 0, \\ -1, & \text{if } X_i < 0, \end{cases}$$

$$S_a = \frac{1}{n} \sum_{i=1}^{n} |X_i|.$$

The scaling factor $S_a$ ensures that the $\mathcal{L}_2$ loss between the full-precision tensor $X$ and its 1-bit quantized counterpart $X^b$ is minimized, significantly reducing quantization error. In theory, the sign function is non-differentiable and does not produce a gradient during the training process. To overcome this challenge, BNNs Courbariaux et al. (2016) introduce the straight-through estimator

(STE) to approximate the gradient of the sign function during backpropagation:

$$g_X = \frac{\partial \mathcal{L}}{\partial X} = \frac{\partial \mathcal{L}}{\partial X^b} \frac{\partial X^b}{\partial X} \approx \frac{\partial \mathcal{L}}{\partial X^b} = g_{X^b}, \tag{2}$$

where $\mathcal{L}$ represents the loss function. In a linear layer, the floating-point multiplications are replaced by efficient bit-wise operations, and memory storage can be reduced by up to $16\times$ compared to FP16 precision. This is particularly beneficial for reducing the inference cost of LLMs.

Training large language models (LLMs) from scratch is highly resource-intensive, requiring thousands of GPUs for days on end and vast amounts of training data on the order of Tera-level tokens. As a result, most quantized LLMs rely on PTQ Frantar et al. (2022); Lin et al. (2024) or low-rank fine-tuning Dettmers et al. (2023); Li et al. (2023), which can be performed with limited computational resources. However, these methods are usually applied to 4-bit as well as 8-bit model quantization. For 1-bit LLMs, we propose three insights based on the analysis of recent researches:

- **PTQ is much unstable for various models.** BiLLM Huang et al. (2024) and ARB-LLM Li et al. (2024) categorize the weights into salient, non-salient weight and sparse area, and conduct elaborate strategies for mixed-precision and parameter updating for these different areas. However, these PTQ methods are not guaranteed to work for all models due to the severe destruction of weights by 1-bit quantization. And for the latest models, such as QWEN3 Yang et al. (2025), BiLLM can easily lead to accuracy collapse Zheng et al. (2025) or severe accuracy degradation.

- **Training from scratch is less efficient.** Table 1 summarizes the experimental settings for recent 1-bit LLMs. BitNet Wang et al. (2023) and FBI-LLM Ma et al. (2024a) propose to train 1-bit and 1.58-bit from scratch with hundreds of billion tokens. However, training a well-optimized 1.58-bit LLM requires significantly more computational resources than a full-precision LLM of the same size Ma et al. (2025). In contrast, OneBit Xu et al. (2024) and our proposed BinaryLLM, only require less than 20B tokens for 1-bit training based on pre-trained models, which reveals that training from scratch is much less cost-effective.

- **Naive quantization is not friendly to pre-training weights.** As shown in Figure 2, we provide the quantization errors as well as the initial training loss values for different Bit bit-widths. Compared to higher bit-widths, ultra-low bit (1/1.58/2-bit) quantization introduces much large relative errors and initial loss. Notably, the initial training loss of 1-bit reaches more than 10, close to the loss value for training from scratch, which demonstrates that the sign function wreaks havoc on the pre-trained weights at the beginning of training.

To explore the best performance of pure 1-bit compression, this paper focuses on how to inherit the parameters of pre-trained LLMs, reducing quantization error during both forward and backward propagation, rather than relying on mixed-precision. We argue that training from pre-trained large language models can also yield high-quality 1-bit LLMs with limited computational resources, thus making better use of the open-source pre-trained models available on platforms like HuggingFace Mohamed Mekkouri & Wolf (2024).

## 3.2 Consistent Progressive Training

The most critical challenge of binary networks training lies in how to converting full-precision parameters to 1-bit values. Unlike randomly initialized parameters, those in pre-trained models have been trained to convergence with large-scale training data, meaning that $\Delta W \approx 0$ Ouyang et al. (2024). As a result, they are highly sensitive to numerical variations. When we apply the naive binarization function, as in Equation 1, to compress the pre-trained parameters at the beginning of training, the initial loss is substantial, and the original convergent state is severely disrupted. To address this issue, a finer-grained quantization progressive strategy is required, where 1-bit LLMs are gradually obtained from pre-trained LLMs.

Ideally, as shown in the middle of Figure 1, we aim to design a progressive approximation function $\mathcal{F}(x, t)$. The initial form ($t \to 0$) of the function is a linear transformation, which helps preserve the convergence status of the original parameters. The final form ($t \to \infty$) is the Sign function, which

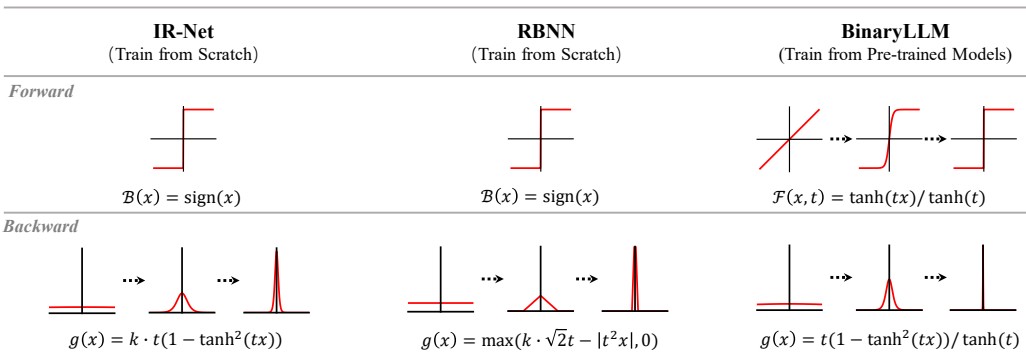

Figure 3: Comparison of different progressive training in binary neural networks.

fully converts the weights into their binarized state. The formula is expressed as follows.

$$\lim_{t \to 0} \mathcal{F}(x, t) \approx x,$$
$$\lim_{t \to \infty} \mathcal{F}(x, t) \approx \text{Sign}(x). \tag{3}$$

With this progressive function $\mathcal{F}(x, t)$, the pre-trained full-precision weights can be smoothly converted to binarized weights throughout the training process. Intuitively, the hyperbolic tangent function $\tanh(x)$ is a suitable choice because it is smooth and has strong linear properties around $x = 0$, while tending to $\pm 1$ for large values of $x$, approximating the $\text{Sign}(x)$ function. Thus we define the progressive function as following:

$$\mathcal{F}(x, t) = \tanh(tx)/\tanh(t) \tag{4}$$

To normalize the function and maintain consistency in scaling, we introduce $\tanh(t)$ in the denominator. This ensures that the function behaves similarly across different stages while controlling the transition rate. The gradient of $\mathcal{F}(x, t)$ *w.r.t* the input $x$ can be obtained by

$$\mathcal{F}'(x, t) = \frac{\partial \mathcal{F}(x, t)}{\partial x} = t(1 - \tanh^2(tx))/\tanh(t) \tag{5}$$

As shown in Figure 3, we visualize the forward and backward of $\mathcal{F}(x, t)$, this function provides a smooth transition between full-precision values and their corresponding 1-bit states, help effectively reducing quantization error and preserving valuable information. Additionally, the parameter $t$ is a scheduler based on training stage, which will be further discussed in the Appendix.

Above all, the final binarized function in 1-bit training is defined as follows:

$$W^b = S_a \times \mathcal{F}(W/S_a, t), \tag{6}$$

where $S_a = \frac{1}{n} \sum_{i=1}^{n} |W_i|$. In the backward, we use the gradient of $\mathcal{F}(x, t)$, which is consistent with the forward propagation instead of STE.

**Comparison with existing progressive training.** Progressive training is a common optimization technique in quantization, yet it has not been explored in LLMs. Unlike CNNs, the pre-trained parameters of LLMs contain rich and effective information due to training on tera-token corpora, making it crucial to prevent weight corruption. As shown in Figure 3, prior progressive methods such as IR-Net Qin et al. (2020) and RBNN Lin et al. (2020) mainly focus on gradient approximation while still relying on the Sign function in the forward pass. This mismatch between forward and backward introduces an optimization gap. Moreover, as discussed earlier, applying naive 1-bit quantization to pre-trained weights causes irreversible damage at the start of training. In contrast, our progressive training maintains consistency between forward and backward: the initial state of $\mathcal{F}(x, t)$ is nearly linear, preserving the weight distribution, and gradually transforms the weights into their binarized states. At convergence, inference is performed with the Sign function instead of $\mathcal{F}(x, t)$, incurring negligible error.

### 3.3 BINARY-AWARE INITIALIZATION

To reduce the difficulty of progressive training, we further propose to preprocess the original weights. In the field of quantized LLMs Lin et al. (2024), it is commonly acknowledged that *weights are not equally important.* We typically categorize weights into salient and non-salient groups based on their sensitivity to quantization. Salient weights, though fewer in number, are much harder to quantize and have a significant impact on the final accuracy, even in 4-bit and 8-bit quantization. Reducing the accuracy degradation caused by salient weights is one of the key challenges in LLM quantization.

GPTQ Frantar et al. (2022) and AWQ Lin et al. (2024) are among the most widely used and efficient post-training quantization methods for weight compression, particularly for bit-widths larger than 2 bits. GPTQ quantizes one group of weights at a time while updating the remaining full-precision weights to compensate for quantization errors. However, this compensation in GPTQ becomes significantly less accurate for ultra-low bit quantization, especially at 1-bit, and it irreversibly disrupts the distribution of the pre-trained weights. In contrast, AWQ introduces an equivalent transformation layer by layer, fusing the magnitude of salient weights with the inputs $A$. The scaling factors are obtained by minimizing the distance between the outputs of the full-precision linear transformation and the binarized linear transformation, as shown in Equation 7.

$$S_t^* = \arg\min_{S_t} \|\mathcal{B}(W \cdot S_t^{-1})(S_t \cdot A) - WA\|, \tag{7}$$

where $S_t$ represents the scaling factors along the input channels. Unlike GPTQ, the equivalent transformation in AWQ does not disturb the original pre-trained weights. However, 1-bit quantization of all linear layers in the model introduces significant quantization errors to the features. This error accumulates progressively from the early layers to the final layers. The layer-wise search method in AWQ does not account for the error accumulation across layers in ultra-low bit quantization, leading to suboptimal performance in the final model.

Therefore, we propose a novel binary-aware initialization that considers error accumulation between different layers, while avoiding irreversible corruption of the original pre-trained weights, the formulation is as follows:

$$S_t^* = \arg\min_{S_t} \sum_i^L \log(p(A_i|A_{i-L}, \ldots, A_{i-1}); \mathcal{B}(W \cdot S_t^{-1})), \tag{8}$$

where $L$ represents the token length, and the conditional probability $p$ follows the standard autoregressive calculation in GPT-style language models with binarized weights. For simplicity, we omit the full representation of weights and intermediate feature transformations. In our approach, the pre-trained weights are frozen, and only the scaling factors $S_t$ are updated during searching. Since the number of scaling factors is relatively small, the entire optimization process requires only a few dozen search steps.

### 3.4 DUAL-SCALING COMPENSATION

In addition to preprocessing pre-trained weights for better initialization, there remains a significant error between $W$ and $W_b$ during 1-bit progressive training in Equation 6. To reduce binarized error, most methods adopt a scaling compensation scheme, as in Equation 1, used by XNOR-Net Rastegari et al. (2016), which minimizes the $\mathcal{L}_2$ distance between full-precision and binarized weights. Inspired by Liu et al. (2020); Tu et al. (2022), we hypothesize that introducing additional learnable parameters can improve 1-bit LLM training. To avoid excessive overhead that could hinder the acceleration and memory efficiency of binary inference, we propose adding extra scaling factors with the same shape as $S_a$, which remains efficient and effective.

FBI-LLM Ma et al. (2024a) is the first to propose assigning a learnable scale to each row or column, initializing them with the value of $S_a$ from Equation 1. However, when applied to 1-bit training with pre-trained LLMs, we observe a significant drop in accuracy. Our analysis suggests that $S_a$ is crucial as it serves as an analytical solution that minimizes the $\mathcal{L}_2$ distance between full-precision and binarized weights at each training step, especially for 1-bit training with pre-trained LLMs. When scaling factors are set as learnable parameters, they are updated along with the loss, which compromises the ability to maintain minimal quantization error at each step, leading to instability and accuracy degradation. To address this, we propose a dual-scaling compensation scheme for the

Table 2: Main results on different evaluation benchmarks. We report the perplexity on Wiki2, C4 and PTB, and zero-shot accuracy on 7 downstream tasks. The results of OneBit Xu et al. (2024), BitNet Ma et al. (2024b) and FBI-LLM Ma et al. (2024a) are from the paper or evaluated on the open-source pre-trained weights.

| Model | Size | Bit | Perplexity ↓ | | | | Zero-shot Accuracy ↑ | | | | | | | |
| | | | Wiki2 | C4 | PTB | Avg | BoolQ | PIQA | HS | WG | ARC-e | ARC-c | OBQA | Avg |
|---|---|---|---|---|---|---|---|---|---|---|---|---|---|---|
| OPT Allal et al. (2024) | 125M | 16.0 | 27.7 | 26.6 | 39.0 | 31.1 | 55.4 | 62.0 | 31.3 | 50.3 | 40.0 | 22.8 | 28.0 | 41.4 |
| SmolLM Allal et al. (2024) | 135M | 16.0 | 17.6 | 22.1 | 36.3 | 25.3 | 60.2 | 68.2 | 42.6 | 53.0 | 56.3 | 28.8 | 34.4 | 49.1 |
| Pythia Biderman et al. (2023) | 160M | 16.0 | 26.6 | 28.9 | 45.5 | 33.6 | 51.3 | 62.1 | 31.3 | 49.6 | 39.1 | 24.0 | 26.8 | 40.6 |
| FBI-LLM Ma et al. (2024a) | 130M | 1.01 | 28.2 | 26.9 | 136.6 | 63.9 | 62.1 | 59.3 | 28.7 | 51.0 | 34.9 | 20.5 | 26.4 | 40.4 |
| **BinaryLLM (Ours)** | 135M | 1.01 | 26.8 | 28.1 | 51.1 | 35.3 | 60.4 | 62.0 | 31.2 | 51.8 | 41.5 | 24.2 | 28.6 | 42.8 |
| OPT Zhang et al. (2022) | 1.3B | 16.0 | 14.6 | 16.1 | 20.3 | 17.0 | 57.8 | 72.5 | 53.7 | 59.5 | 51.0 | 29.5 | 33.4 | 51.1 |
| LLaMA3 Dubey et al. (2024) | 1.3B | 16.0 | 9.8 | 14.0 | 17.6 | 13.8 | 64.0 | 74.5 | 63.7 | 60.5 | 60.4 | 36.4 | 26.6 | 55.2 |
| BitNet Ma et al. (2024b) | 1.3B | 1.59 | 24.1 | 21.8 | 145.1 | 63.7 | 56.7 | 68.8 | 37.7 | 55.8 | 54.9 | 24.2 | 19.6 | 45.4 |
| OneBit-OPT Xu et al. (2024) | 1.3B | 1.02 | 25.4 | 23.0 | - | - | 59.5 | 62.6 | 34.3 | 51.1 | 41.3 | 24.1 | - | - |
| FBI-LLM Ma et al. (2024a) | 1.3B | 1.01 | 12.6 | 13.8 | 39.3 | 21.9 | 60.3 | 69.0 | 42.3 | 54.0 | 43.6 | 25.3 | 29.6 | 46.3 |
| **BinaryLLM (Ours)** | 1.3B | 1.01 | 14.7 | 18.4 | 27.0 | 20.0 | 60.4 | 70.0 | 49.6 | 56.4 | 47.4 | 26.0 | 30.8 | 48.7 |
| OPT Zhang et al. (2022) | 2.7B | 16.0 | 12.5 | 14.3 | 18.0 | 14.9 | 60.4 | 74.8 | 60.6 | 61.0 | 54.4 | 31.3 | 35.2 | 54.0 |
| Pythia Biderman et al. (2023) | 2.8B | 16.0 | 10.2 | 14.6 | 18.3 | 14.4 | 64.7 | 73.7 | 59.3 | 60.1 | 58.8 | 33.0 | 35.6 | 55.0 |
| LLaMA3 Dubey et al. (2024) | 3.0B | 16.0 | 7.8 | 13.5 | 11.3 | 10.9 | 73.4 | 77.5 | 73.6 | 69.9 | 71.6 | 46.0 | 43.0 | 65.0 |
| OneBit-OPT Xu et al. (2024) | 2.7B | 1.02 | 21.9 | 20.8 | - | - | 54.3 | 63.9 | 38.2 | 51.7 | 43.4 | 24.4 | - | - |
| BitNet Ma et al. (2024b) | 3.0B | 1.59 | 10.0 | 9.8 | 85.0 | 34.9 | 61.5 | 71.5 | 42.9 | 59.3 | 61.4 | 28.3 | 61.5 | 50.2 |
| **BinaryLLM (Ours)** | 3.0B | 1.01 | 12.4 | 17.1 | 20.4 | 16.6 | 62.2 | 72.7 | 58.3 | 66.4 | 63.1 | 34.4 | 42.0 | 57.0 |

forward propagation in the 1-bit training stage, retaining both $S_a$ and $S_l$. The updated formula is as follows:

$$W^b = S_l \times S_a \times \mathcal{F}(W/S_a, t), \tag{9}$$

where all $S_l$ are initially set to 1 and then updated along with the 1-bit weights during training. The first scaling factor, $S_a$, minimizes the majority of quantization error, while the second scaling factor, $S_l$, focuses on compensating for accuracy at each training step. In the inference stage, the two scaling factors can be merged into a single one without introducing any additional computational overhead, and the progressive function is replace with Sign function:

$$W^b = S \times \text{Sign}(W), \tag{10}$$

where $S = S_l \times S_a$. As shown in Figure 1, the 1-bit inference of our proposed BinaryLLM adopts a simple approach and does not introduce extra offsets in each row or column.

## 4 EXPERIMENTS

In this section, we demonstrate the performance of our proposed BinaryLLM via comparisons with existing 1-bit LLMs and extensive ablation experiments.

### 4.1 EXPERIMENTAL SETUP

**Benchmarks.** We evaluate the performance of our proposed BinaryLLM and existing 1-bit LLMs on two different metrics: perplexity and zero-shot accuracy. Following the setting and library version with FBI-LLM Ma et al. (2024a), we conduct the evaluation on *lm-evaluation-harness* EleutherAI (2021). We test the perplexity on Wiki2 Merity et al. (2016), C4 Raffel et al. (2020) and PTB Melis et al. (2017), which is better when the value is lower. And we test the zero-shot accuracy on 7 downstream tasks, including BoolQ Clark et al. (2019), PIQA Bisk et al. (2020), HellaSwag Zellers et al. (2019), Winogrande Sakaguchi et al. (2021), ARC Clark et al. (2018), and OpenbookQA Mihaylov et al. (2018), higher accuracy value represents better performance.

**Baselines.** We conduct experiments on LLMs with different model sizes, from smallest to largest being 130M, 1.3B, and 3B. The full-precision baselines we choose are SmolLM-135M Allal et al. (2024), LLaMA3-1B (actually 1.26B) Dubey et al. (2024) and LLaMA3-3B Dubey et al. (2024). We also compare our proposed BinaryLLM with the state-of-the-art 1-bit and 1.58-bit works, such as OneBit Xu et al. (2024), BitNet Wang et al. (2023) and FBI-LLM Ma et al. (2024a). For some ablation and toy experiments that validate the effect of different techniques, we use the Pythia-70M Biderman et al. (2023).

Table 3: Comparison results on different evaluation benchmarks. We report the perplexity on Wiki2, C4 and PTB, and zero-shot accuracy on 7 downstream tasks.

| Model | Method | Perplexity ↓ | | | | Zero-shot Accuracy ↑ | | | | | | | |
|---|---|---|---|---|---|---|---|---|---|---|---|---|---|
| | | Wiki2 | C4 | PTB | Avg | BoolQ | PIQA | HS | WG | ARC-e | ARC-c | OBQA | Avg |
| Pythia-160M | FP16 | 26.6 | 28.9 | 45.5 | 33.6 | 51.3 | 62.1 | 31.3 | 49.6 | 39.1 | 24.0 | 26.8 | 40.6 |
| Biderman et al. (2023) | BinaryLLM | 35.4 | 33.2 | 53.1 | 40.6 | 58.5 | 60.3 | 29.4 | 52.3 | 37.8 | 22.6 | 25.2 | 40.9 |
| | Δ | +8.8 | +4.3 | +7.6 | +7.0 | +7.2 | -1.8 | -1.9 | +2.7 | -1.3 | -1.4 | -1.6 | +0.3 |
| SmolLM 135M | FP16 | 17.6 | 22.1 | 36.3 | 25.3 | 60.2 | 68.2 | 42.6 | 53.0 | 56.3 | 28.8 | 34.4 | 49.1 |
| Allal et al. (2024) | BinaryLLM | 25.9 | 27.0 | 47.6 | 33.5 | 61.6 | 60.9 | 30.7 | 50.4 | 40.9 | 23.1 | 28.4 | 42.3 |
| | Δ | +8.3 | +4.9 | +11.3 | +8.2 | +1.4 | -7.3 | -11.9 | -2.6 | -15.4 | -5.7 | -4.0 | -6.7 |

**Implementation Details.** In this paper, the training data is randomly sampled from RedPajama dataset Weber et al. (2024). Our BinaryLLMs of different model sizes are constructed based on the open-source pre-trained models SmolLM-135M Allal et al. (2024), LLaMA3-1B (actually 1.26B) Dubey et al. (2024) and LLaMA3-3B Allal et al. (2024). For toy experiments and ablations, we sample 50 billion tokens for Pythia-70M. During training, the sequence length is set to 2048, the batch size is 128 and the optimizer we adopt is AdamW. The learning rate decreases from $1e-4$ to $2e-6$ with cosine scheduler, and the weight decay is set to 0.1. The details are shown in Appendix.

## 4.2 MAIN RESULTS

Table 2 summarizes the primary results comparing our proposed BinaryLLM against multiple state-of-the-art baseline models. Since BitNet uses a higher quantization precision, we report the average bit of all models for the sake of clarity of exposition. Following FBI-LLM, we calculate the average bit-width on the linear layers of all the transformer blocks, while neither the embedding layer nor the output header is taken into account. For fair comparison, we only list the 1-bit LLMs from quantization-aware training for that the 1-bit LLMs from post-training quantization is much worse without abundant training. BinaryLLM achieves the state-of-the-art performance compared with other 1-bit LLMs on various model sizes.

For the 130M-scale comparison, BinaryLLM is trained on the pre-trained SmolLM Allal et al. (2024), and only takes 20 billion tokens in the 1-bit training process. The results shows that BinaryLLM outperforms the 1-bit FBI-LLM, which is training from scratch on 1.26 T tokens, including Wiki2, PTB and five zero-shot tasks. The average perplexity of BinaryLLM is 28.6 higher than that of FBI-LLM, and the average accuracy is 2.4% higher on the zero-shot tasks. Although our method drops 10.0 and 6.3% in perplexity and zero-shot accuracy, respectively, compared to SmolLM-135M, this result is still close to the full-precision OPT-125M and Pythia-160M.

When conducting 1-bit compression on 1.3B-scale large language models, we choose the latest LLaMA3-2-1B, which is pruned from larger model and apply knowledge distillation to recover performance. Compared with BitNet and FBI-LLM, BinaryLLM get much better performance on both perplexity and zero-shot accuracy. For a better comparison, we also list the results of OneBit Xu et al. (2024) and its corresponding full-precision models OPT Zhang et al. (2022). Our BinaryLLM drops 4.9 and 4.4 perplexity on both Wiki2 and C4, however, OneBit drops 10.8 and 6.9 on these two test sets, receptively, much worse than our methods.

For the larger scale on 3B LLMs, we further train our BinaryLLM on LLaMA3-3B, which are much resource-intensive. Compared to the full-precision models, BinaryLLM drops 5.7 and 7.0% on average perplexity and zero-shot accuracy. The results are also very competitive with BitNet-1.58bit of 3B model size, although the model performs very well on Wiki2 and C4. In addition, we also find that perplexity of the 1-bit LLMs trained from scratch is much poor on the PTB task, while the LLMs trained from the pre-training model are balanced on each task. This phenomenon further indicates that the quantized LLM trained from pre-training weights has more potential for development of ultra-low bit quantization.

In the future, we will further explore the possibility of obtaining high-precision 1-bit LLMs on pre-trained large-scale language models using lower training data and training cost

Table 4: Comparison results on different evaluation benchmarks. We report the perplexity on Wiki2, C4 and PTB, and zero-shot accuracy on 7 downstream tasks.

| Method | Perplexity ↓ | | | | Zero-shot Accuracy ↑ | | | | | | | |
|---|---|---|---|---|---|---|---|---|---|---|---|---|
| | Wiki2 | C4 | PTB | Avg | BoolQ | PIQA | HS | WG | ARC-e | ARC-c | OBQA | Avg |
| FP16 | 17.6 | 22.1 | 36.3 | 25.3 | 60.2 | 68.2 | 42.6 | 53.0 | 56.3 | 28.8 | 34.4 | 49.1 |
| vanilla 1-bit | 30.4 | 31.1 | 52.9 | 38.1 | 58.3 | 59.0 | 29.3 | 51.5 | 37.2 | 22.8 | 28.6 | 41.0 |
| IR-Net Qin et al. (2020) | 29.8 | 30.7 | 51.1 | 37.3 | 59.2 | 59.5 | 30.3 | 50.3 | 37.3 | 23.1 | 28.3 | 41.1 |
| BinaryLLM | 27.4 | 28.2 | 49.3 | 35.0 | 61.0 | 60.6 | 30.8 | 50.5 | 40.3 | 22.9 | 28.1 | 42.0 |

Table 5: Perplexity of different scaling factors on Pythia-70M Biderman et al. (2023). The base method refers to using only $S_a$ as defined in Equation 1, while FBI-LLM represents the use of $S_l$, initialized with $S_a$. Our proposed method is shown in Equation 9.

| Methods | Perplexity ↓ | | | |
|---|---|---|---|---|
| | **Wiki2** | **C4** | **PTB** | **Average** |
| $S_a$ | 73.1 | 54.8 | 98.7 | 75.5 |
| $S_l$ | 76.2 | 58.5 | 96.4 | 77.0 |
| $S_l \times S_a$ | 68.7 | 51.5 | 92.5 | 70.9 |

### 4.3 ABLATION STUDIES

**1-bit LLMs from Different Baselines.** Since our BinaryLLM is initialized from a pre-trained full-precision model, we examine how its accuracy influences the final 1-bit model. Experiments on Pythia-160M Biderman et al. (2023), under the same settings as SmolLM-135M Allal et al. (2024), show that despite Pythia having 25M more parameters, both models are comparable in scale. As shown in Table 3, the average perplexity degradation is similar, but stronger full-precision models suffer sharper drops on certain metrics (e.g., SmolLM drops 11.3 on PTB vs. 7.6 for Pythia). For zero-shot accuracy, Pythia's weak full-precision baseline allows its 1-bit version to match or exceed it, while SmolLM shows consistent accuracy loss across tasks, with a 6.7 average drop. Overall, better-trained models exhibit greater degradation after 1-bit quantization, consistent with the intuition that denser knowledge is harder to preserve. Nonetheless, we recommend starting from advanced models, as their 1-bit versions still outperform those derived from weaker models. Future work will focus on mitigating this degradation.

**Effectiveness on Progressive Training.** As shown in Table 4, when we apply progressive of IR-Net Qin et al. (2020) on vanilla methods, perplexity and zero-shot accuracy improve slightly. When we conduct consistency on vanilla methods, the perplexity drop 4.6 and zero-shot accuracy improve 1.3%. Consistent progressive training can greatly improve the stability of 1-bit LLM training, whereas naive 1-bit quantization of the pre-trained weights leads to very large initial loss and worse performance.

**Effectiveness on Scaling Factors.** As shown in Table 5, using the learnable scaling factors $S_l$, initialized with $S_a$, results in much higher perplexity on datasets like Wiki2, C4, and PTB compared to using the fixed, non-learnable $S_a$. And we evaluate our proposed dual-scaling compensation scheme, and the results demonstrate that maintaining both the original analytical solution $S_a$ and the additional learnable parameter $S_L$ leads to significant improvements in accuracy.

## 5 CONCLUSION

In this paper, we present a new paradigm for training 1-bit LLMs by leveraging pre-trained models, addressing the limitations of prior methods that train from scratch. To bridge the large gap between full-precision and binary parameters, we propose consistent progressive training, enabling a smooth transition in both forward and backward passes. To further boost performance, we introduce binary-aware initialization to preserve salient weights and dual-scaling compensation to balance error minimization and accuracy. Experiments on LLMs of various sizes show that our approach achieves state-of-the-art results among 1-bit LLMs and greatly narrows the gap to full-precision models. These findings demonstrate the feasibility of high-performance 1-bit LLMs without costly retraining, paving the way for efficient and scalable deployment.

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

## A    TRAINING PIPELINES

Actually, there are two stages in the training of our proposed BinaryLLM:

**Stage1: Binary-Aware Initialization**. We introduce per-input channel-wise scaling factor $S_t$ to each linear layer of transformer blocks, aiming at preserving the salient weights before the binary training. In practical, we scale the weights along the input channel ($W \cdot S_t^{-1}$) and inversely scale the input tensor ($A \cdot S_t$) at the same time. During the search, all the elements of scaling factor $S_t$ are initialized to 1, and other parameters, such as weights in RMSNorm layers and linear layers, are frozen. To obtain a binary-friendly scale, we conduct 1-bit quantization on the scaled weight. As shown in Equation 11, we adopt an end-to-end manner to search the optimal scaling factors with autoregressive loss. Because only the parameters of $S_t$ participate in the training, the whole training takes only 50 steps with a few minutes, which is much efficient. Then we get the scaled weights $\widetilde{W}$ for the progressive training of the second stage.

$$S_t^* = \arg\min_{S_t} \sum_i^L \log(p(A_i | A_{i-L}, \ldots, A_{i-1}); \mathcal{B}(W \cdot S_t^{-1})),$$

$$\widetilde{W} = W/S_t^*. \tag{11}$$

**Stage2: Progressive Training**. In the second stage, the entire 20B training data is evenly divided into 20 chunks, and the entire training period is also divided into 20 phases, where each chunk is trained in corresponding phase. With dual-scaling compensation and progressive training, we quantize the weights as Equation 12.

$$\widetilde{W}^b = S_l \times S_a \times \mathcal{F}(\widetilde{W}/S_a, t), \tag{12}$$

where

$$S_a = \frac{1}{n} \sum_{i=1}^n |\widetilde{W}_i|,$$

$$\mathcal{F}(x, t) = \texttt{tanh}(tx)/\texttt{tanh}(t).$$

During training, all parameters are learnable and the hyperparameter $t$ changes along the stage. We recommend using the exponential progressive scheduler $t(c) = 1.3 \cdot e^{0.22c} - 1.3$, where c denotes chunk number.

## B    COMPARISON WITH 1-BIT PTQ METHODS

Although 1-bit post-training quantization methods are not the primary baselines for comparison in our work, we nevertheless applied the open-source implementations of BiLLM Huang et al. (2024) and ARB-LLM Li et al. (2024) to perform 1-bit quantization on LLaMA3-1B and LLaMA3-3B models. As shown in Table 6, we evaluate the perplexities of Wikitext2 and C4 for full-precision models, BiLLM, ARB-LLM and our proposed BinaryLLM. We can clearly observe that PPL of BiLLM and ARB-LLM drops significantly compared to the full-precision models, with several metrics exceeding 100, rendering the models nearly unusable. In contrast, our method, which employs quantization-aware training, is able to maintain relatively strong performance metrics despite the 1-bit quantization.

## C    SCALING UP TO 7B MODELS

We further extend our method to larger-scale model LLaMA2-7B, to validate the scalability and effectiveness of our proposed scheme. As shown in Table 7, our proposed BinaryLLM, significantly outperforms existing baselines such as OneBit Xu et al. (2024) and FBI-LLM Ma et al. (2024a) in terms of average perplexity on WikiText2 and C4, with improvements of 1.2 and 1.1, respectively. Furthermore, in zero-shot evaluation, BinaryLLM achieves an average score of 56.7, outperforming FBI-LLM by 3.8, and reducing the gap to the full-precision LLaMA2 model to 10. Due to limited computational resources, we are unable to extend our experiments to larger models such as LLaMA3-8B or LLaMA2-13B. Nevertheless, we believe the current experimental results provide strong evidence of the effectiveness and scalability of our proposed BinaryLLM.

Table 6: Perplexities of different 1-bit methods on LLaMA3 Dubey et al. (2024).

| Models | Methods | Bit | Perplexity ↓ | | |
|--------|---------|-----|------|------|---------|
| | | | Wiki2 | C4 | Average |
| **LLaMA3-1B** Dubey et al. (2024) | Full-Precision | 16 | 17.6 | 22.1 | 19.9 |
| | BiLLM Huang et al. (2024) | 1.08 | 815.4 | 610.2 | 712.8 |
| | ARB-LLM-X Li et al. (2024) | 1.08 | 115.4 | 144.0 | 129.7 |
| | BinaryLLM | 1.01 | **14.7** | **18.4** | **16.6** |
| **LLaMA3-3B** Dubey et al. (2024) | Full-Precision | 16 | 7.8 | 13.5 | 10.7 |
| | BiLLM Huang et al. (2024) | 1.08 | 104.3 | 82.8 | 93.6 |
| | ARB-LLM-X Li et al. (2024) | 1.08 | 49.2 | 56.6 | 52.9 |
| | BinaryLLM | 1.01 | **12.4** | **17.1** | **14.8** |

Table 7: Main results on different evaluation benchmarks. We report the perplexity on Wiki2, C4 and PTB, and zero-shot accuracy on 7 downstream tasks.

| Model | Size | Bit | Perplexity ↓ | | | Zero-shot Accuracy ↑ | | | | | | | |
|-------|------|-----|------|------|-----|------|------|------|------|-------|-------|------|------|
| | | | Wiki2 | C4 | Avg | BoolQ | PIQA | HS | WG | ARC-e | ARC-c | OBQA | Avg |
| LLaMA2 | 7B | 16 | 5.5 | 7.3 | 6.4 | 77.7 | 79.1 | 76.0 | 69.1 | 74.6 | 46.2 | 44.2 | 66.7 |
| OneBit-LLaMA2 | 7B | 1.01 | 9.7 | 11.1 | 10.4 | 63.1 | 68.1 | 52.6 | 58.4 | 41.6 | 29.6 | - | - |
| FBI-LLM | 7B | 1.01 | 9.1 | 10.5 | 10.3 | 61.5 | 72.6 | 57.7 | 58.9 | 53.0 | 29.9 | 36.8 | 52.9 |
| **BinaryLLM (Ours)** | 7B | 1.01 | 8.7 | 9.7 | 9.2 | 65.5 | 73.2 | 59.5 | 66.4 | 60.3 | 37.2 | 34.5 | 56.7 |

## D  EFFICIENCY ANALYSIS

In this section, we present an efficiency analysis of different quantization methods for LLMs.

We evaluate the memory footprint and the theoretical number of computation cycles for various quantization methods based on LLaMA2-7B. To estimate memory, we compute the effective bit of the Linear layers based on their average bit and assume other parameters are stored in 16-bit precision. For the number of cycles, we adopt a theoretical model in which the number of cycles required for a matrix multiplication is determined by the bitwidths of its two inputs. Specifically, the cycle count for multiplying matrix $A$ and $B$ is computed as $\text{Cycle}(A, B) = \text{Bit}_A \times \text{Bit}_B$, which help us compare the bit-level computational cost under a unified theoretical framework. As shown in Table 8, 1-bit quantization achieves approximately $10.06\times$ memory compression and $10.12\times$ theoretical speedup compared to full-precision models. This highlights the significant potential of extreme quantization in reducing both memory footprint and computational cost for large-scale LLMs.

In addition, we conduct latency evaluations on real-world hardware to assess the practical efficiency of low-bit quantization. We adopt the T-MAC Wei et al. (2024) framework, a CPU-based inference engine for low-bit LLM deployment on Edge, which supports various bit-width configurations. Due to the difficulty of deploying true 1-bit quantization, we follow the BitNet-style format for 1-bit inference. Our experiments is carried out on an AMD EPYC 7642 48-Core Processor. As shown in Table 4, we report the inference latency of the LLaMA2-7B model under different quantization settings (4-bit-g128, 2-bit-g128, and 1-bit with channel-wise quantization) and different thread number. The results demonstrate that 1-bit quantization yields significant latency improvements. However, since the actual bit-width used in deployment is 1.58-bits, the gains over 2-bit are not yet substantial. We believe that with the hardware optimization for 1-bit quantization in the future, Binarized LLMs remains promising for efficient LLM inference.

## E  EFFECTIVENESS ON BINARY-AWARE INITIALIZATION

As shown in Figure 5, we conduct comparison experiments on different pre-processing methods. We observe that the weights processed using AWQ, GPTQ, or a combination of AWQ+GPTQ perform much worse than the base scheme, which involves no preprocessing. In contrast, our proposed binary-aware initialization (BaI) significantly reduces the initial training loss to 7.2 and lowers the

| Models | Average Bit | Memory (GB) | Cycles (T) |
|---|---|---|---|
| LLaMA2-7B | 16.0 | 13.48 | 1.72 |
| LLaMA2-7B-INT4 | 4.0 | 3.76 | 0.48 |
| BitNet | 1.58 | 1.80 | 0.23 |
| BiLLM | 1.08 | 1.40 | 0.18 |
| OneBit | 1.01 | 1.34 | 0.17 |
| FBI-LLM | 1.01 | 1.34 | 0.17 |
| BinaryLLM | 1.01 | 1.34 | 0.17 |

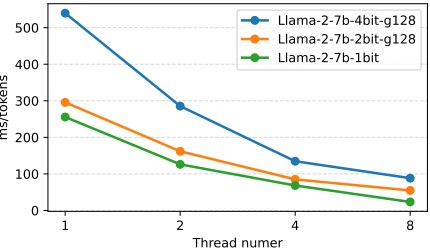

Table 8: Memory and cycles of different methods.

Figure 4: Latency with different threads.

perplexity by an order of magnitude. Furthermore, we conducted an ablation study on binary-aware initialization using the LLaMA-1B model. As shown in Table 9, applying our proposed initialization leads to consistent improvements across the WikiText-2, C4, and PTB datasets. Although the gains are relatively modest—likely due to the limited number of salient weights in smaller models, we believe this technique will have a more pronounced impact when scaling to larger models.

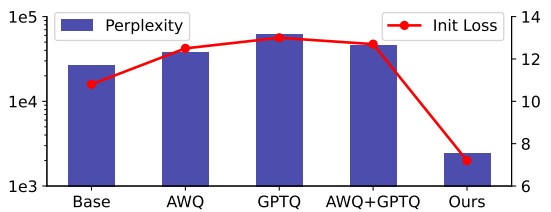

| Models | Wiki2 | C4 | PTB | Avg. PPL |
|---|---|---|---|---|
| FP Model | 9.8 | 14.0 | 17.6 | 13.8 |
| w/ BaI | 14.7 | 18.4 | 27.0 | 20.0 |
| w/o BaI | 15.1 | 18.7 | 27.5 | 20.4 |

Figure 5: Comparison of initial perplexity and loss.

Table 9: Abaltion on initilization.

## F  PROGRESSIVE SCHEDULER

Progressive scheduler function $t = \phi(c)$ determines the transition rate from $y = x$ to $y = \texttt{sign}(x)$ during the entire training stages. We explore fours schemes: uniform progressive, logarithm progressive, exponential progressive and degree uniform progressive. Figure 6 shows different formula of scheduler function, As we can see that, when $t$ varies uniformly across training phases, the corresponding approximation curves does not vary uniformly, sparse in the beginning and dense at the final, which is also similar with logarithm progressive. Due to the fast change in the initial stage, the information of original pre-trained weight cannot be effectively retained. leading a bad performance on the final 1-bit LLMs. In contrast, the approximate curves obtained by exponential progressive and degree uniform progressive vary more smoothly. The former generates $t$ with a exponential manner and the later makes the degree of the angle between the gradient direction and the y axis of the different curves at the (0,0) position vary evenly. They all got much better performance on the final 1-bit LLMs, and exponential progressive achieve better performance for that they pay more attention on the later stages, which makes the approximation of $y = \texttt{sign}(x)$ much better. Finally, we believe there exists a better progressive scheduler function that could further boost the 1-bit LLMs from pre-trained models,we are willing to discuss this with the community.

## G  VISUALIZATION OF WEIGHTS

To further investigate the behavior of our proposed progrssive training, we visualize the weight distribution of the layers.0.mlp.down_proj layer from LLaMA3-1B. As shown in Figure 7, the weights progressively evolve towards a 1-bit state as training progresses, driven by the effect of the progressive quantization function as Equation 12. This observation aligns well with our expectation that the model will gradually adapt to 1-bit representations during training.

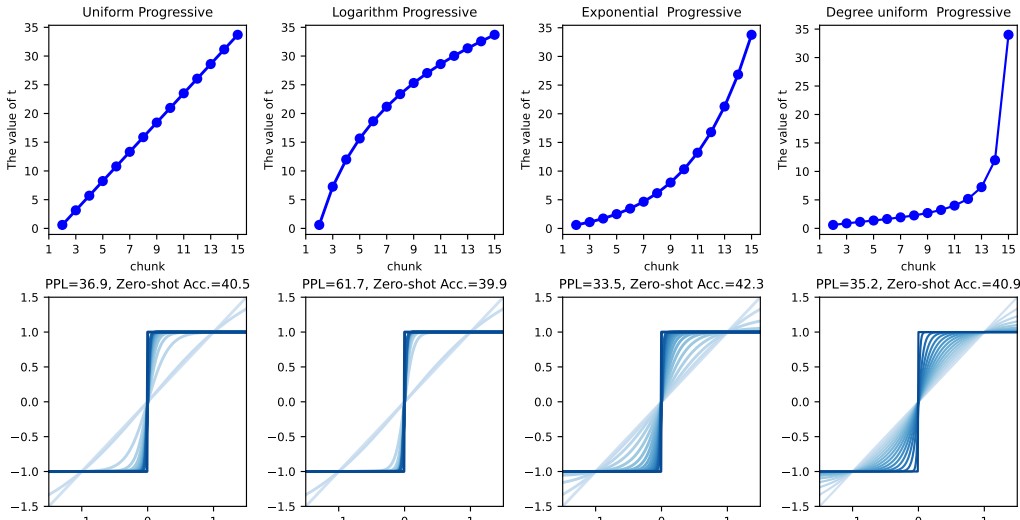

Figure 6: Different progressive scheduler on $t$. The top and bottom of each column represents the function of $t$ and the corresponding progressive approximation curves. We also list the perplexity and zero-shot accuracy of final 1-bit models.

# H  THE USE OF LARGE LANGUAGE MODELS (LLMS)

In preparing this paper, we made limited use of a large language model (LLM) to aid in improving the clarity and readability of the manuscript. Specifically, the LLM was employed for language polishing, including grammar correction and stylistic refinement. The LLM was not involved in research ideation, problem formulation, methodology design, experimental execution, or result analysis. All technical content, experiments, and conclusions are entirely the work of the authors.

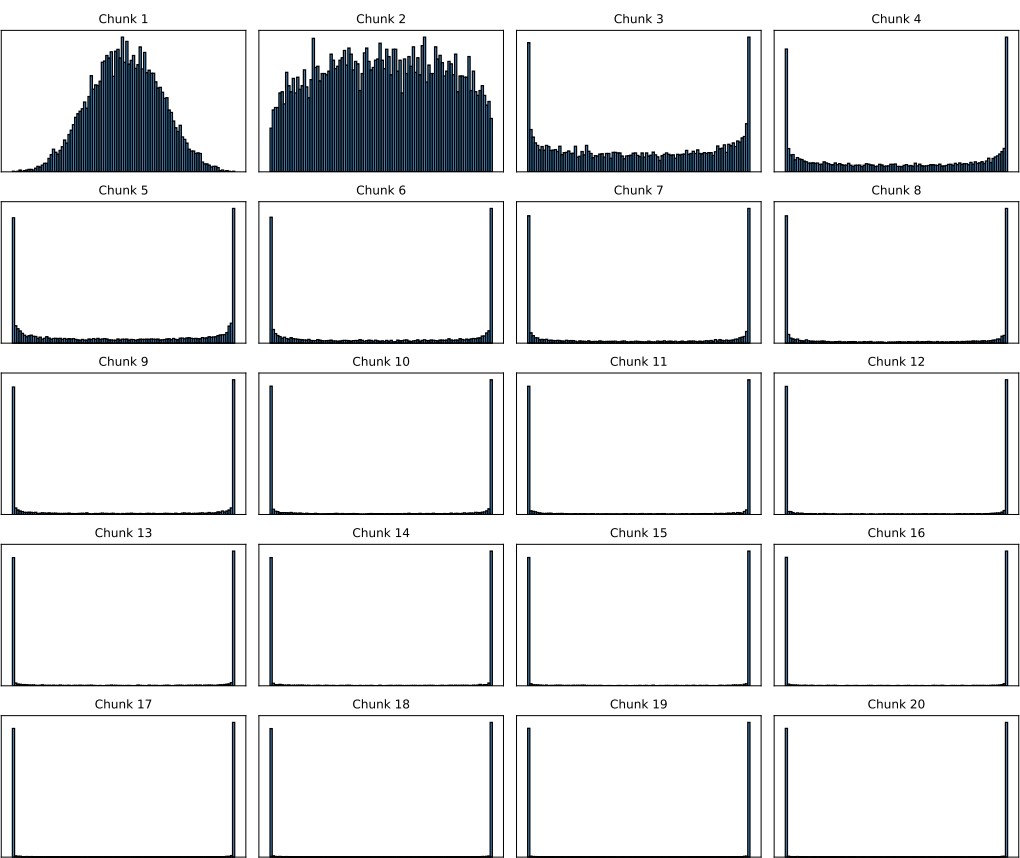

Figure 7: Weights distribution of different training chunks.

