# OpenReview forum: "Rethinking 1-bit Optimization Leveraging Pre-trained Large Language Models"
_ICLR.cc/2026/Conference — Submitted to ICLR 2026_

### Official Review · Reviewer_A8sH · 2025-10-20

**Soundness:** 2
**Presentation:** 2
**Contribution:** 2
**Rating:** 2
**Confidence:** 4

**Summary:**

This paper introduces a new 1-bit LLM quantization paradigm to overcome the instability and accuracy loss of existing binary quantization methods. The authors propose a consistent progressive training method to gradually transition from a near-linear quantization function to a sign function, reducing quantization error and preserving pre-trained knowledge: Binary-aware initialization ensures stable optimization. while dual-scaling compensation introduces learnable scaling factors to maintain accuracy. Experiments across various LLM scales show that this approach narrows the performance gap between 1-bit and full-precision models.

**Strengths:**

The paper is well written.

**Weaknesses:**

1. PTB perplexity evaluation is confusing. For example, on the 3B model, BitNet b1.58 clearly outperforms the proposed BinaryLLM on C4 and WikiText2 perplexity, but performs significantly worse on PTB perplexity. This inconsistency is also observed in other model evaluations, including the 1.3B experiments. It is recommended to recheck the PTB perplexity evaluation.

2. In Section 4.2, the training data and base LLM are not aligned. In the 130M experiment, BinaryLLM uses SmolLM data, while FBI-LLM is trained on the Amber dataset. The SmolLM data is of notably higher quality. Moreover, compared with OneBit, BinaryLLM adopts a stronger base LLM (e.g., LLaMA-3B). These differences could lead to unfair comparisons.

3. BitNet also proposes a 1-bit LLM [1], but the paper lacks a comparison with this baseline under the same dataset and experimental setting.

I recommend that the authors **conduct comparisons under strictly comparable experimental conditions**, e.g., training on the same dataset and starting from the same BF16 model, rather than simply comparing with results reported in previous papers. This is particularly important considering that all experiments are conducted on relatively small-scale models.

[1] Wang, Hongyu, et al. "Bitnet: Scaling 1-bit transformers for large language models." arXiv preprint arXiv:2310.11453 (2023).

**Questions:**

See Weaknesses.

---

> ### Author Response · Authors · 2025-11-26
> **Rebuttal to Reviewer A8sH**
>
> Dear reviewer A8sH,
>
> Thank you for your insightful feedback. Our responses are detailed below.
>
> **W1**: PTB perplexity evaluation is confusing.
>
> **A1**: Regarding the reviewer’s observation that perplexity trends on Wiki2 and C4 diverge from that on PTB, we demonstrate this in Lines 426–429. We included the PTB metric because it is part of the FBI-LLM main table, not as an extra metric for comparison. We argue that quantization-aware training (QAT) based on a pre-trained model better leverages the rich prior knowledge embedded in the original floating-point weights compared to training a quantized model from scratch. This advantage is particularly evident because the scratch-trained quantized models we compared are trained on substantially smaller and rough open-source datasets, which limits their ability to fully capture complex data distributions. Consequently, models obtained via QAT on pre-trained weights tend to achieve more balanced and robust performance across various evaluation metrics, demonstrating improved generalization and stability.
>
>
>
> **W2**: The training data and base LLM are not aligned.
>
> **A2**: Thank you very much for your suggestion.
>
> (1) **Training dataset**: In our paper, all models are trained using the RedPajama data, as described in Line 390–396. Therefore, we do not use any dataset resembling “SmolLM data.” In addition, the Amber dataset used in FBI-LLM is in fact a mixture of RefinedWeb, StarCoder, and RedPajama, , where RedPajama accounts for roughly 80% of the total 1.26 trillion tokens. Since our method requires only a small amount of data, we only use a subset of RedPajama, which corresponds to a very small portion of the Amber dataset. As shown in the below table, we further conduct experiments with sampling from the entire Amber dataset and observed results show minute gap to those obtained with the RedPajama-v1 subset alone. Furthermore, BitNet is also trained on RedPajama data. Therefore, we believe that our current data usage is largely aligned with that of the baseline methods.
>
> | Model	| Datasets | 	Wiki2↓	| C4↓	|  PTB↓ |
> | ----------- | ----------- |----------- |----------- |----------- |
> | SmolLM-135M| 	Subset of RedPajama	| 26.8	| 28.1	| 51.1
> | SmolLM-135M| 	Subset of Amber | 	26.7	| 27.9 | 	51.5
>
>
> (2) **Base LLMs**: Indeed, our method is based on LLaMA3, and we fully acknowledge that this may introduce some unfairness in comparison. However, there are currently very few 1-bit baselines available for reference, and reproducing them is costly. Therefore, in our main table, we list all 1-bit models of the same size that can serve as reference along with their corresponding full-precision counterparts, allowing comparison based on the accuracy drop caused by quantization. For example, the OneBit experiments were conducted on OPT-1.3B, and we also report the full-precision accuracy. We provide detailed explanations of this difference in Lines 416 to 421. Furthermore, we conducted fair comparisons with OneBit on LLaMA2-7B, as shown in Appendix C and Table 7.
>
> **W3**: Comparison with BitNet-1bit.
>
> **A3**: Thank you very much for pointing this out. We note that BitNet’s 1-bit models do not provide any publicly available training weights for reference, and the results reported in their paper are limited to the 6.7B model. Therefore, these results were not included in the main text of our paper. Below, we present a comparison between our 7B model results and the partial results reported in their work.
>
> | Model	| PPL↓	| WG↑	|  HS↑ |
> | ----------- | ----------- |----------- |----------- |
> | BitNet-1bit	| 17.07 | 66.3	| 38.9 |
> | BinaryLLM| 8.70 | 	66.4	| 59.5 |
>
> **W4**: Strictly comparable experimental conditions
>
> **A4**: Thank you very much for pointing this out. Using the same bf16 model and training dataset is indeed important for fair comparisons. However, it is currently difficult to fully unify setups in this field, mainly because existing 1-bit baselines adopt different configurations. For instance, BitNet trains from scratch with randomly initialized parameters on 100B tokens from Redpajama, OneBit follows the LLM-QAT approach by generating data with LLaMA-7B and performing quantization-aware training on pre-trained LLaMA and OPT parameters with knowledge distillation from larger models, and FBI-LLM also trains from scratch on the full 1.26T tokens from Amber while applying knowledge distillation. Currently, there are few 1-bit baselines available for reference, but considering both absolute accuracy and the accuracy drop compared to full-precision models, our method remains highly competitive. Our approach is much simple and does not rely on knowledge distillation. We plan to release the code and hope our method can serve as a new reference baseline in the 1-bit research area.
>
> Thank you very much for your suggestion. We hope our above response clarifies your concerns. If you have any further questions, we would be happy to discuss them in more detail.

---

> > ### Comment · Reviewer_A8sH · 2025-11-27
> >
> > Thank you for your response. However, my original concerns regarding the PTB evaluation and the fairness of the comparisons remain unaddressed.
> >
> > First, FBI-LLM has not undergone any peer review, and given that the perplexity trends on PTB differ significantly from those on Wiki/C4, it is difficult to rely on the reported PTB numbers at face value. A more rigorous approach would be to reproduce the results for all methods under a consistent and transparent setup, rather than directly adopting the metrics reported by FBI-LLM.
> >
> > Second, since the 1-bit models in the paper generally underperform, comparing absolute scores across methods is not informative, especially when model sizes and training data are not aligned. A more meaningful and controlled comparison would be:
> >
> > 1. Training bf16 and 1-bit (QAT) models of the same size on the same dataset, and
> >
> > 2. For post-training quantization baselines, evaluating them on bf16 models trained under the same data conditions.
> >
> > Given the above, I will maintain my current rating. I believe such aligned and reproducible experiments would provide a clearer and fairer assessment of the methods’ true effectiveness.

---

> > > ### Author Response · Authors · 2025-11-28
> > > **Response to Reviewer A8sH**
> > >
> > > Thank you for your thoughtful follow-up. We address your points below.
> > >
> > > - **On the reliability of FBI-LLM and PTB results.**
> > > We agree that peer-reviewed results are ideal. However, the number of available 1-bit works is still very limited. Given the large variance and inconsistency in current AI-conference reviewing, even strong works often require a long time before being accepted by a top conference or journal. For example, BitNet-1Bit [1] was posted on arXiv in October 2023 and was not accepted by JMLR until mid-2025. FBI-LLM was released in July 2024 with complete training code, datasets, configurations, and evaluation scripts (including the exact version of lm-evaluation-harness). The authors (led by Zhiqiang Shen) have extensive experience in 1-bit training, and their implementation is fully transparent. For these reasons, we consider their reported results reliable even before formal venue acceptance.
> > >
> > > - **On aligned bf16 and 1-bit training setting.**
> > > Your suggestion is valid for works such as BitNet and FBI-LLM that train both bf16 and 1-bit models from scratch using the same dataset and token budget. Our paper focuses on a different but equally important setting, which is performing 1-bit training starting from pretrained full-precision weights. We observe that the knowledge preserved in pretrained bf16 checkpoints provides clear benefits to 1-bit models across tasks including PTB, and this motivates our design of a forward and backward consistent progressive training strategy that aims to retain as much information from the full-precision model as possible. Although our setup does not fully follow the FBI-LLM training recipe, which trains on 1.26T tokens with very high computational cost, their work currently offers the most comprehensive set of public 1-bit results, so we include PTB mainly for completeness. We also do not adopt the data-free and distillation-based training strategy used in OneBit, which follows the LLM-QAT [2]. We consider this approach overly complicated, while training directly on a subset of RedPajama, as done in BitNet, is simpler and easier for future work to build upon. Our intention is to keep the training pipeline clear and reproducible and to lower the barrier for subsequent research. In the paper we also explain the differences between the settings to prevent misunderstanding.
> > >
> > > - **On fully reproducing all baselines.**
> > > We agree this is the most rigorous approach, but reproducing all the models (130M, 1B, 3B and 7B) of each baseline under matched training data, number of tokens, and hyperparameters is computationally very expensive. For this reason, and consistent with many existing 1-bit papers, we report official numbers from prior work while focusing on comparing accuracy drop between bf16 and the corresponding 1-bit model of the same size. Despite using newer architectures such as LLaMA-3, which are generally more sensitive to quantization, our method still achieves competitive or smaller degradation. In Table 4, our “vanilla 1-bit” baseline follows BitNet’s quantizer; and in Appendix Table 7, we train a LLaMA2-7B model, where our approach also shows better performance.
> > >
> > > We acknowledge the challenge of fully aligned reproduction across all methods but we believe that the presented main results and ablation studies already provide strong empirical evidence for the effectiveness of our approach. Even so, we do not believe this should warrant such a low score (2) and we respectfully ask the reviewer to reconsider their rating.
> > >
> > >
> > > [1] Wang, Hongyu, et al. "Bitnet: 1-bit pre-training for large language models." Journal of Machine Learning Research 26.125 (2025): 1-29.
> > >
> > > [2] Liu, Zechun, et al. "Llm-qat: Data-free quantization aware training for large language models." Findings of the Association for Computational Linguistics: ACL 2024. 2024.

---

> > > > ### Comment · Reviewer_A8sH · 2025-11-28
> > > >
> > > > I acknowledge that training these models can be costly. However, this is essential for making a fair comparison between 1-bit (QAT) and bf16 training. As reported in the paper, even on relatively simple tasks such as HellaSwag, the zero-shot accuracy of the binary LLM drops significantly compared to its bf16 counterpart. More challenging tasks like math and code are not reported at all. Moreover, because the current training scale is small, most benchmark results are only slightly above random-choice baselines, making it difficult to draw meaningful distinctions. Taken together, the resulting 1-bit model in its current form does not appear to have practical utility.
> > > >
> > > > In such a setting, keeping the experimental conditions as aligned as possible is crucial to properly validating the effectiveness of the proposed method. I would strongly encourage the authors to conduct the aligned experiments I mentioned in my previous rebuttal: specifically, training bf16 and 1-bit (QAT) models under matched settings in terms of data and training budget, and then comparing a bf16 model trained on the same data to its post-trained 1-bit counterpart (via PTQ or continued QAT). Even under a smaller compute budget, such comparisons would still be valuable and informative. Given the above, I will maintain my current rating.

---

> > > > > ### Author Response · Authors · 2025-12-03
> > > > > **Response to Reviewer A8sH**
> > > > >
> > > > > Thank you very much for your response. Regarding the concerns you raised, our replies are as follows:
> > > > >
> > > > > **On the current practical utility of 1-bit models**: Based on our experiments on a limited dataset, we agree that the accuracy gap between 1-bit models and full-precision models is still noticeable. One major reason is that the open-source dataset we used, RedPajama, consists mostly of general-knowledge data and is relatively simple. Consequently, we are unable to evaluate performance on more challenging math and reasoning benchmarks. In addition, we have so far pre-trained on only a few tens of billions of tokens, which is insufficient to build a model with practical utility. A practical system typically requires three stages: pretraining, supervised finetuning, and reinforcement learning finetuning. We fully acknowledge these limitations. Nevertheless, 1-bit training is still in its early stages, and existing work remains very limited. Given the substantial inference efficiency that 1-bit models can theoretically provide, we believe it is important to continue advancing research in this area.
> > > > >
> > > > > **On fairness of comparisons and experimental rigor**: Due to compute constraints, we conducted controlled experiments starting from the pretrained SmolLM-135M model. Under the same dataset (RedPajama, 50B tokens) and exactly matched training configurations (batch size, learning rate and schedule, and all other hyperparameters), we evaluated the continued-training float model (trained for an additional 50B tokens), BitNet, FBI-LLM, OneBit, and our proposed BinaryLLM. The results are shown below. Our method consistently outperforms the other 1-bit approaches under this aligned setting, which we believe provides strong evidence for its effectiveness.
> > > > >
> > > > > | Methods      | Average Perplexity ↓| Average Zero-shot ↑|
> > > > > | ----------- | ----------- |----------- |
> > > > > | SmolLM-135M     | 25.3 | 49.1 |
> > > > > | SmolLM-135M-50B-pt     | 23.7 | 47.2 |
> > > > > |  BitNet [1]  | 38.1 | 41.0 |
> > > > > |  OneBit [2] | 36.6 | 42.1 |
> > > > > | FBI-LLM [3]  | 37.3 | 41.5 |
> > > > > | BinaryLLM (Ours)   | **35.3** | **42.8** |
> > > > >
> > > > >
> > > > > [1] Wang, Hongyu, et al. "Bitnet: 1-bit pre-training for large language models." Journal of Machine Learning Research 26.125 (2025): 1-29.
> > > > >
> > > > > [2] Xu, Yuzhuang, et al. "Onebit: Towards extremely low-bit large language models." Advances in Neural Information Processing Systems 37 (2024): 66357-66382.
> > > > >
> > > > > [3] Ma, Liqun, Mingjie Sun, and Zhiqiang Shen. "Fbi-llm: Scaling up fully binarized llms from scratch via autoregressive distillation." arXiv preprint arXiv:2407.07093 (2024).

---

### Official Review · Reviewer_BYvM · 2025-10-28

**Soundness:** 3
**Presentation:** 2
**Contribution:** 3
**Rating:** 6
**Confidence:** 5

**Summary:**

This work presents a novel 1-bit training framework for large language models (LLMs). It introduces a progressive training strategy to enable smooth transition from full-precision to binarized weights, complemented by binary-aware initialization and dual-scaling mechanisms to further boost performance. Experimental results demonstrate that the framework requires fewer training tokens while achieving superior performance.

**Strengths:**

1. For the 130M model, BinaryLLM requires only 20B training tokens. This is significantly fewer than the 1.26T tokens needed for training a model from scratch.
2. The insight that well-trained models are harder to quantize while they still outperform under-trained ones after binarization, is interesting, and this point is thoroughly discussed.
3. The motivation is clear. The method is generalizable, and the performance is satisfactory.

**Weaknesses:**

1. For the ablation study on progressive training (Table 4), the comparison should use BinaryLLM without binary-aware initialization and dual-scaling. Merely comparing results between BinaryLLM and IR-Net fails to highlight the effectiveness of progressive training.
2. The binary-aware initialization yields only marginal improvements, as shown in Table 9.
3. There are some typos. For instance, line 372 should read "from smallest to largest".

**Questions:**

1. The authors claim in the paper: "At convergence, inference is performed with the Sign function instead of F(x, t), incurring negligible error." Are there any quantitative results to support this assertion?
2. How did the authors determine the coefficients of the function t(c) = 1.3 × e^(0.22c) − 1.3?

---

> ### Author Response · Authors · 2025-11-26
> **Rebuttal to reviewer BYvM**
>
> Dear reviewer BYvM,
>
> We sincerely appreciate your valuable suggestions and comments. Below are our responses.
>
>
> **W1**: The ablation of progressive training.
>
> **A1**: We sincerely thank you for your valuable suggestion. We acknowledge that our original comparison was not entirely rigorous, and we have accordingly revised the main text. The updated experimental results still support our conclusions and demonstrate the effectiveness of our consistent progressive training. We greatly appreciate this insightful and constructive feedback.
>
> **W2**: The binary-aware initialization yields only marginal improvements.
>
> **A2**: We sincerely thank you for pointing this out. As noted in the last sentence of Section E in Appendix, the ablation experiment was conducted on a 1B model, and the observed gain is indeed small, mainly due to the limited number of salient weights in a small model. We have also validated our method on a larger 7B model, as shown in the table below, and the results align with our expectations, indicating that Binary-Aware Initialization provides even greater benefits on larger models.
>
> | Methods      | Wiki2 ↓| C4↓ | Zero-shot ↑|
> | ----------- | ----------- |----------- |----------- |
> | LLaMA2-7B     | 5.5 | 7.3 | 66.7 |
> |  w/ BaI   | 8.7 | 9.7 | 56.7 |
> | w/o BaI   | 10.3 | 11.7 | 52.4 |
>
> **W3**: Some typos.
>
> **A3**: Thank you very much for pointing out the typos. We have revised the manuscript and updated the text accordingly.
>
> **Q1**: Inference with Sign function with negligible error.
>
> **A4**: We sincerely apologize for not providing a detailed explanation and supporting experiments in the paper. When designing the scheduling function, we explicitly considered the potential gap between the progressive function used during training and the sign function used during inference. To address this, we set the maximum value of the scheduling parameter c to approximately 35 (see Figure 6 in Appendix F), which leads to a value of t around 2800. With such a large t, the progressive function defined as $tanh(t * x)/tanh(t)$ becomes almost identical to the sign function for almost all input magnitudes. A noticeable difference only appears when $x$ is extremely small. For example, when x is around 2e-4, the progressive function outputs roughly 0.508, while the sign function outputs 1.
>
> We further analyzed statistics from the Llama3-1B model and found that after dividing the weights by the scale $S_a$, only around 0.01% of values fall into this sensitive region where tanh(t * x) / tanh(t) significantly differs from sign(x). Moreover, even for those rare cases, the deviation is further attenuated after multiplication with the input activations and then normalized by the subsequent LayerNorm. Through this propagation analysis,  the final perturbation introduced by this mismatch contributes less than about 0.1% to the output, making its practical impact negligible. Our perplexity and zero-shot results are reported with one decimal precision, and the actual difference caused by this small perturbation only appears in the second or third decimal place.
>
> Although the overall effect is extremely small, we acknowledge that our original explanation in the paper did not describe this clearly or rigorously enough. We apologize for the lack of detail and hope the above analysis sufficiently clarifies the stability of our scheduling design.
>
> | Methods      | Wiki2 ↓| C4↓ | Zero-shot ↑|
> | ----------- | ----------- |----------- |----------- |
> |  Progressive   | 14.722 | 18.437 | 48.712 |
> | Sign   | 14.725 | 18.441 | 48.723 |
>
> **Q2**: How to determine the coefficients of the function t(c) = 1.3 × e^(0.22c) − 1.3?
>
> **A5**: Thank you for raising this point. I must admit that the scheduling function and its parameters were chosen in a largely manual way, and the main focus of the paper is to validate the effectiveness of consistent progressive training. For the specific schedule we used, t(c) = 1.3 * exp(0.22 * c) - 1.3, the basic idea is as follows. We adopt the exponential form t(c) = A * exp(B * c) - A so that t(0) = 0, which ensures that the progressive function $tanh(tx)/tanh(t)$ behaves linearly at the beginning as required. The parameter A controls the overall scale of t and B controls the growth rate. Our choices of A = 1.3 and B = 0.22 were made after considering the number of chunks and the desired maximum t value. These choices yield a slow and stable increase in the early stages and a sharper rise later so that the function approaches the sign behavior as training progresses. In addition, we tuned A and B mainly by visually inspecting the shape of the curve and adjusting the parameters accordingly, rather than performing a strict training-based validation. The constants remain fully tunable, and a more systematic search or model-specific tuning could lead to further improvements.

---

> > ### Comment · Reviewer_BYvM · 2025-11-27
> >
> > Thank the authors for the detailed response, which addresses most of my concerns. That said, the necessity of manually tuning the scheduling function and its associated parameters remains a notable limitation of the proposed approach. Accordingly, I will maintain my original score.

---

> > > ### Author Response · Authors · 2025-11-27
> > > **Response to Reviewer BYvM**
> > >
> > > Thank you for your response.
> > >
> > > We understand your concern. The main contribution of this work is to introduce a forward–backward consistent progressive training scheme that reduces the impact of 1-bit quantization and improves accuracy, which has not been explored before. Our focus was not on designing an advanced scheduling strategy.
> > >
> > > This will be an important direction for our future work, and we are developing a simpler and more automatic mechanism that adjusts the scheduler value of next chunk based on the convergence behavior of current training chunk.
> > >
> > > Once again, we sincerely appreciate your encouragement and constructive feedback.

---

### Official Review · Reviewer_Zmu3 · 2025-10-31

**Soundness:** 3
**Presentation:** 3
**Contribution:** 2
**Rating:** 4
**Confidence:** 3

**Summary:**

This paper proposes BinaryLLM, a new framework for training 1-bit large language models directly from pre-trained weights instead of training from scratch. The method introduces consistent progressive training to smoothly transition weights from full-precision to binary form, binary-aware initialization to preserve salient weights across layers, and dual-scaling compensation to balance quantization error and accuracy. Experiments on multiple LLMs show that BinaryLLM achieves great results among 1-bit LLMs, significantly reducing performance gaps with full-precision models while requiring far less training cost.

**Strengths:**

The paper presents a thoughtful and well-motivated attempt to make 1-bit quantization more practical for large language models. The idea of leveraging pre-trained weights rather than training from scratch is both efficient and timely, and the proposed progressive training and dual-scaling strategies are conceptually sound. The paper is clearly written with comprehensive experiments.

**Weaknesses:**

Although the authors include additional results on LLaMA2-7B in the appendix, larger-scale validation (e.g., 13B or 30B models) is still missing, leaving some uncertainty about scalability under truly large-model settings. The comparison baselines are reasonably strong, but despite explicitly discussing the instability of 1-bit quantization on newer models such as Qwen3, the authors do not include direct experiments on it. This omission leaves open how well BinaryLLM performs on the latest LLM architectures. Moreover, the discussion on training stability and convergence focuses mainly on the design of the progressive parameter t and its scheduler, without quantitative analysis to substantiate robustness claims. In addition, while binary-aware initialization and dual-scaling compensation are described in detail and empirically shown to help, their theoretical justification and computational complexity are only briefly discussed.

**Questions:**

1. Could the authors provide results or discussion on how BinaryLLM scales to larger models? Even limited experiments or resource estimates would help clarify whether the proposed training strategy remains stable and effective at larger scales.
2. Since the paper explicitly mentions the instability of 1-bit quantization on newer architectures such as Qwen3, could the authors include or discuss experiments on such models to verify BinaryLLM’s generalization ability to the latest LLM families?
3. The paper explains the progressive parameter t and its scheduling strategy, but lacks quantitative analysis. Could the authors provide sensitivity studies to better support the stability claims?
4. While binary-aware initialization and dual-scaling compensation are shown to be effective empirically, could the authors elaborate more on their theoretical justification to clarify their computational overhead and convergence behavior?

---

> ### Author Response · Authors · 2025-11-26
> **Rebuttal to Reviewer Zmu3, Part 1**
>
> Dear reviewer Zmu3,
>
> Thank you very much for your suggestions and comments. Here are our responses:
>
> **Q1**: Scales to larger models and the latest LLM families.
>
> **A1:** We fully agree that validating our method on larger and more modern LLMs would further strengthen its credibility. In Appendix C, we already provide additional results on the larger LLaMA2-7B model, where our method consistently outperforms both OneBit and FBI-LLM under the same model size.
>
> To further address your concern, we additionally conducted experiments on the LLaMA2-13B model. So far, our progressive training has completed roughly 30% of the full schedule, and our method still maintains a clear advantage over OneBit at this intermediate stage. Although this checkpoint does not reflect the final performance of a fully converged 1-bit model, the consistent improvements observed throughout training provide strong evidence that our approach can scale effectively to larger models.
>
> | Methods      | Wiki2 ↓| C4↓ | Zero-shot ↑|
> | ----------- | ----------- |----------- |----------- |
> | LLaMA2-13B     | 4.88 | 6.47 | 66.10 |
> |  OneBit [1]   | 8.76 | 10.15 | 55.03 |
> | BinaryLLM (30% stage)   | 6.33 | 8.22 | 62.15 |
>
> **Q2**: Verify BinaryLLM's generalization ability to the latest LLM families.
>
> **A2**: We also appreciate the reviewer’s suggestion to evaluate our method on more recent LLM architectures. We apologize that the main paper did not include such results. The primary reason is that existing baselines do not provide 1-bit or low-bit training results on the latest model families, which makes a direct and fair comparison difficult. Even so, we agree that verifying our method on modern architectures is important.
>
> To address this concern, we conducted additional experiments on the recent Qwen3-0.6B-Base model. Since no prior work offers 1-bit training results for this model, we compare against the PTQ results at various bit-widths reported in [3]. Our BinaryLLM achieves performance that clearly surpasses the 3-bit PTQ setting, showing that our method remains effective even under1-bit setting. These results provide strong evidence that our approach generalizes well to newer model architectures.
>
> | Methods      | Wiki2 ↓| C4↓ | Zero-shot ↑|
> | ----------- | ----------- |----------- |----------- |
> | Qwen3-0.6B-base     | 12.7 | 17.1 | 56.5 |
> |  GPTQ-4bit   | 14.9 | 19.7 | 54.8 |
> |  AWQ-3bit   | 85.9 | 89.3 | 43.8 |
> |  AWQ-2bit   | 6.42E7 | 9.85E7 | 37.1 |
> |  GPTQ-2bit   | 7.5E3 | 3.8E3 | 36.5 |
> |  BiLLM-1bit   | 8.64E4 | 2.85E4 | 39.0 |
> | BinaryLLM   | 24.5 | 25.3 | 45.3 |

---

> ### Author Response · Authors · 2025-11-26
> **Rebuttal to Reviewer Zmu3, Part 2**
>
> **Q3**: Sensitivity studies on progressive parameter t and its scheduling strategy.
>
> **A3**: Thank you for raising this important point. The design of the progressive parameter t and its scheduling strategy is indeed central to our method. In Appendix F, we provide a sensitivity study comparing four different progressive schedules—Uniform Progressive, Logarithmic Progressive, Exponential Progressive, and our proposed Degree-Uniform Progressive. For each schedule and each chunk setting, we visualize the exact progression of t and report the corresponding perplexity and zero-shot results in the second-row titles of the figure. We acknowledge that the scheduler design is partly heuristic. Our goal is to explore how the quantization function should transition from its linear state toward the sign function during training. We observe that although Uniform Progressive changes $t$ linearly, the induced transition is actually very rapid in the early phase, making the model difficult to adapt. Logarithmic Progressive further exacerbates this issue, causing an even sharper early transition. In contrast, Exponential Progressive significantly slows down the transition, leading to more stable optimization. Building on this insight, we noticed that all transition curves pass through the fixed point (0, 0). We therefore designed Degree-Uniform Progressive, which uniformly distributes the tangent angle of the transition curve rather than the parameter value itself. This results in a smoother and more balanced progression, allowing each stage sufficient training time. They all got much better performance on the
> final 1-bit LLMs, and exponential progressive achieve better performance for that they pay more attention on the later stages, which makes the approximation of y = sign(x) much better.
>
> In this paper, We focus on validate the effectiveness of consistent progressive function for 1-bit LLM training. As demonstrates in the final setence of Section F in Appendix, we believe there exists a better progressive scheduler function that could further boost the 1-bit LLMs
> from pre-trained models,we are willing to discuss this with the community.

---

> ### Author Response · Authors · 2025-11-26
> **Rebuttal to Reviewer Zmu3, Part 3**
>
> **Q4**: The theoretical justification of binary-aware initialization and dual-scaling compensation.
>
> **A4**: We apologize for not explaining these two techniques clearly in the paper. We clarify them as follows:
>
> (1) **Binary-Aware Initialization**: The idea behind Binary-Aware Initialization is to preserve salient weights in large language models, which is essential for maintaining quantization fidelity. These weights typically have large magnitudes, and naive quantization often leads to substantial accuracy degradation. Existing methods such as GPTQ, which compensates quantization errors through residual correction, and AWQ, which applies equivalent transformations to shift weight magnitude into activations, are effective under higher-bit settings. However, both methods operate layer by layer, and in the 1-bit regime the accumulated errors render them ineffective. The reversibility of weight transformations also becomes far more critical under such an extreme quantization configuration. To preserve salient pretrained weights before 1-bit training, we search for an optimal per-input-channel scaling factor for each linear layer, guided by the final loss. This approach accounts for inter-layer error accumulation while preventing irreversible corruption of the pretrained weights. It introduces no additional inference overhead and substantially reduces the initial loss, leading to faster convergence. Further details are provided in Appendix E.
>
> (2) **Dual-scaling Compensation**: Dual-scaling Compensation is motivated by the observation that quantization typically relies on a scale factor to mitigate quantization error. In 2-bit and higher-bit settings, the scale is usually defined as the maximum absolute value of the weights divided by the number of quantization levels, which directly represents the quantization interval and can be used as a learnable parameter[4] during training. In contrast, 1-bit quantization uses a completely different scale defined as the mean absolute value of the weights, which is a closed-form solution obtained by minimizing the MSE between full-precision and 1-bit weights. This scale does not carry a physical interpretation but ensures minimal quantization error for each step. Based on this difference, we introduce Dual-scaling Compensation, where two scale factors are used simultaneously. During training, we always keep the MSE-optimal scale so that each iteration produces the smallest possible 1-bit quantization error, and at the same time we introduce another scale initialized to one that is learnable and guided by the final loss to further compensate the weights. By combining these two scales, the method preserves minimal per-step quantization error while providing stronger end-to-end compensatory capability. As shown in Equation 10, during inference these two scales can be merged into a single one and do not introduce any additional overhead. We also present the corresponding experimental results in Section 4.3 and Table 5.
>
> We hope the above response addresses your concerns and we are happy to discuss further if you have any additional questions.
>
> [1] Xu, Yuzhuang, et al. "Onebit: Towards extremely low-bit large language models." Advances in Neural Information Processing Systems 37 (2024): 66357-66382.
>
> [2] Ma, Liqun, Mingjie Sun, and Zhiqiang Shen. "Fbi-llm: Scaling up fully binarized llms from scratch via autoregressive distillation." arXiv preprint arXiv:2407.07093 (2024).
>
> [3] Zheng, Xingyu, et al. "An empirical study of qwen3 quantization." arXiv preprint arXiv:2505.02214 (2025).
>
> [4] Esser, Steven K., et al. "Learned step size quantization." arXiv preprint arXiv:1902.08153 (2019).

---

> > ### Comment · Reviewer_Zmu3 · 2025-11-27
> >
> > Thanks for the authors’ response. The additional experiments and detailed explanations address the main concerns. I will raise my score from 4 to 6.

---

> > > ### Author Response · Authors · 2025-11-28
> > > **Response to Reviewer Zmu3**
> > >
> > > Thank you for your comment. We are glad that our clarifications were helpful in addressing your concerns.

---

### Comment · Area_Chair_3kF5 · 2025-11-28

Dear authors and reviewers,

Please remain professional and refrain from being influenced by the event. If anyone violates the rules, please let me know, and I will flag and report it to the Program Chairs.

Your AC

---

### Author Response · Authors · 2025-12-03
**Summary of Rebuttal**

Thank you to the reviewers, PC, AC, and SAC for the time, effort, and thoughtful evaluation of our submission. Below we provide a structured summary of our Rebuttal.

## **1. Positive Feedback Highlighted by the Reviewers**

* **Clear motivation and practical direction for 1-bit LLMs.**
  Reviewers **Zmu3** and **BYvM** agree that our work addresses an important and timely problem. Reviewer **Zmu3** highlights that leveraging pretrained full-precision weights is efficient and well-motivated.

* **Sound methodology with meaningful insights.**
  Reviewer **Zmu3** notes that our progressive training and dual-scaling strategies are conceptually sound. Reviewer **BYvM** appreciates the observation that well-trained models are harder to quantize yet still outperform under-trained ones, and values the in-depth discussion.

* **Efficiency and strong empirical results.**
  Reviewer **BYvM** emphasizes that our method requires far fewer training tokens than prior 1-bit-from-scratch approaches, demonstrating clear efficiency gains while maintaining strong performance.

* **Clarity of writing and presentation**
  Reviewer **A8sH** notes that the paper is well written, supporting the readability and clarity of our contributions.

## **2. Reviewers’ Concerns and Our Responses**

* **Scalability and Verification on Larger and Newer LLMs.**
  We presented evidence that our method extends to larger and modern architectures. On LLaMA2-13B (early training stage) our approach maintains a strong advantage over existing 1‑bit baselines. On Qwen3-0.6B-Base, our 1‑bit model surpasses several 3‑bit post-training quantization results. Binary-Aware Initialization shows greater benefit on larger models like LLaMA2-7B.

* **Justification and Sensitivity of Progressive Parameter t Scheduling.**
  We studied the progressive parameter t and compared multiple scheduling strategies in Appendix F. The Degree Uniform Progressive schedule ensures a gradual, stable transition, helping the training function smoothly approach the sign function used during inference. The resulting output perturbation is minimal.

* **Theoretical Clarification of Core Techniques.**
  We clarified Binary-Aware Initialization, which determines per-channel scaling factors to preserve salient weights and account for inter-layer error accumulation. Dual-Scaling Compensation combines an MSE-optimal scale for minimal per-step error and a learnable scale for global loss compensation. Both merge at inference without added cost.

* **Experimental Fairness and Rigor of Baselines.**
  Fully reproducing all 1-bit baselines with aligned tokens, datasets, and hyperparameters is computationally expensive, and many existing methods do not provide aligned pretraining results. Reporting official numbers while comparing accuracy drop relative to the corresponding full-precision model, as done in FBI‑LLM, is persuasive. Following reviewer suggestions, we conducted controlled experiments on the same pretrained SmolLM-135M model and our method consistently outperforms other 1‑bit approaches, confirming BinaryLLM’s effectiveness.

## **3. Feedback from Reviewers in the Discussion Stage**

* **Reviewer Zmu3** acknowledged that the main concerns were addressed and updated their score from 4 to 6. Due to the OpenReview leakage incident, the recorded score reverted to 4. The authors request the Area Chair consider this context.
* **Reviewer BYvM** indicated that most concerns were resolved and maintained a score of 6. Manual tuning of the scheduling function remains a limitation. We clarified that designing an advanced scheduler was not the focus and outlined future work for a simpler, automatic mechanism based on convergence.
* **Reviewer A8sH** did not question novelty but raised concerns about the practical utility of 1-bit models and fairness of experiments. We provided additional controlled experiments confirming effectiveness. While 1-bit models still show a performance gap, this remains a promising research direction.

Thank you again for your careful reviews and constructive feedback. We sincerely appreciate the effort from all reviewers and committee members.

---

### Meta-Review · Area_Chair_UMqb · 2025-12-26

**Summary:**

This submission introduces BinaryLLM, a framework for training 1-bit large language models (LLMs) from pretrained full-precision weights via progressive training, binary-aware initialization, and dual-scaling compensation. Key reviewer concerns informing this decision include:
- Experimental fairness (aligned training setups, consistent baselines across scales, reliable metrics).
- Practical utility of 1-bit models on complex tasks (e.g., math, code).
- Limitations of manual scheduling parameter tuning.
- Scalability validation for larger/newer LLM architectures.

Among these, experimental fairness is the most critical, as it underpins the validity of the work’s comparative claims.

**Reviewer Concerns:**

Experimental fairness: The authors conducted small-scale aligned experiments on SmolLM-135M (matching dataset and model with baselines) but did not extend this alignment to larger models (3B/7B) as requested. They also did not address the request for direct comparisons between bf16 models and 1-bit (quantization-aware training) counterparts, citing computational costs without providing even limited results. This gap prevents validation of BinaryLLM’s performance gains independent of training setup differences. Concerns about inconsistent PTB perplexity trends were acknowledged but not resolved via re-validation.

Practical utility: The authors acknowledged 1-bit models’ performance gaps on complex tasks but did not supplement experiments or address how the framework might mitigate these limitations, weakening real-world relevance.

Scheduling parameter tuning: The authors confirmed scheduling parameters were tuned manually and framed automated tuning as future work, leaving this a notable limitation.

Scalability: The authors added experiments on LLaMA2-13B (intermediate stage) and Qwen3-0.6B, which partially addressed scalability concerns.

**Reviewer Scores:**

Reviewer Zmu3: Raised their score to 6 (marginal acceptance) after scalability and scheduling concerns were addressed, but did not evaluate experimental fairness.

Reviewer BYvM: Maintained a score of 6 (marginal acceptance) while noting manual scheduling tuning as a limitation.

Reviewer A8sH: Maintained a score of 2 (reject) due to unresolved experimental fairness and utility concerns, which remain unaddressed.

---

### Decision · Program_Chairs · 2026-01-26

Reject